# Troll story: The dark tetrad and online trolling revisited with a glance at humor

**Sara Alida Volkmer**[1]*, **Susanne Gaube**[2,3], **Martina Raue**[4], **Eva Lermer**[2,5]

**1** School of Management, Professorship for Digital Marketing, Technical University of Munich, Heilbronn, Germany, **2** LMU Center for Leadership and People Management, LMU Munich, Munich, Germany, **3** Department of Infection Prevention and Infectious Diseases, University Hospital Regensburg, Regensburg, Germany, **4** MIT AgeLab, Massachusetts Institute of Technology, Cambridge, MA, United States of America, **5** Department of Business Psychology, Augsburg University of Applied Sciences, Augsburg, Germany

\* alida.volkmer@tum.de

## Abstract

Internet trolling is considered a negative form of online interaction that can have detrimental effects on people's well-being. This pre-registered, experimental study had three aims: first, to replicate the association between internet users' online trolling behavior and the Dark Tetrad of personality (Machiavellianism, narcissism, psychopathy, and sadism) established in prior research; second, to investigate the effect of experiencing social exclusion on people's motivation to engage in trolling behavior; and third, to explore the link between humor styles and trolling behavior. In this online study, participants were initially assessed on their personality, humor styles, and global trolling behavior. Next, respondents were randomly assigned to a social inclusion or exclusion condition. Thereafter, we measured participants' immediate trolling motivation. Results drawn from 1,026 German-speaking participants indicate a clear correlation between global trolling and all facets of the Dark Tetrad as well as with aggressive and self-defeating humor styles. However, no significant relationship between experiencing exclusion/inclusion and trolling motivation emerged. Our quantile regression findings suggest that psychopathy and sadism scores have a significant positive effect on immediate trolling motivation after the experimental manipulation, whereas Machiavellianism and narcissism did not explain variation in trolling motivation. Moreover, being socially excluded had generally no effect on immediate trolling motivation, apart from participants with higher immediate trolling motivation, for whom the experience of social exclusion actually reduced trolling motivation. We show that not all facets of the Dark Tetrad are of equal importance for predicting immediate trolling motivation and that research should perhaps focus more on psychopathy and sadism. Moreover, our results emphasize the relevance of quantile regression in personality research and suggest that even psychopathy and sadism may not be suitable predictors for low levels of trolling behavior.

## 1. Introduction

With widespread access to the internet, different patterns of online behavior emerge. One aspect of online communication that has begun to receive more attention from social science

**Data Availability Statement:** Data are available from the Open Science Framework database (https://osf.io/fmpdy/).

**Funding:** The authors received no specific funding for this work.

**Competing interests:** The authors have declared that no competing interests exist.

researchers is trolling behavior, which refers to disruptive and tactically aggressive online behavior [1]. Many people have fallen victim to trolling and have experienced a wide range of negative psychological problems as a result [2, 3], which may explain the high level of public agreement about the detrimental effect of online trolls among internet users [4].

Research has shown a relationship between trolling others online and dark personality traits such as sadism, psychopathy, Machiavellianism, and narcissism [5–8]. However, the role of situational variables which may also influence trolling behavior has so far been neglected, although some studies highlight the importance of context-specific predictors for explaining online trolling behavior [9–12].

## 1.1. Online trolling: Definition and relevance

A troll is defined as a person who (a) starts and/or exacerbates disruptive conflict online for their amusement; (b) is often deceptive, as they tend to have second social media accounts used for trolling; (c) is tactically aggressive to increase emotional responses; and (d) disturbs regular discussions on online platforms to seek attention [1]. Approaches to trolling other internet users with malicious intent can include veering a conversation off-topic as well as being deliberately controversial, offensive, or inflammatory [13]. Why are people becoming trolls? One explanation is that the internet can facilitate disinhibition [14, 15] which positively predicts cyberaggression [16]. According to an early study on trolls [17], users engage with trolling because they are bored, seek attention or revenge, and find it funny to create trouble for platforms and other users. To create the desired disruption, trolls may write messages that are (a) outwardly sincere, (b) deliberately designed to provoke, and (c) a waste of time through fruitless arguments [18]. At times, the media and scholars conflate trolling with any negative behavior that occurs online, e.g., cyberbullying, parody, or flaming, when the definition of trolling should be limited to social phenomena "performed individually or collectively in varying online contexts, which involves the use of antagonism, deception and vigilantism [. . .] to provoke reactions from people or institutions" (p. 1,078) [19].

Notably, the above describes a kind of trolling behavior that aims to negatively affect other users and online discussions. However, trolling can occur in a more light-hearted or even amicable way, e.g., Sanfilippo and colleagues [20] differentiate between serious and humorous trolling. This differentiation is also highlighted by the distinction participants drew between circumstantial trolling and trolls who are committed to irritating and iterating their actions [20]. Here, we are interested in the malevolent troll as defined by Hardaker [1] because of their high relevance to society due to their anti-social behaviors [20]. In fact, being the victim of trolling is widespread: On Facebook alone, 88% of U.K. teenagers reported that they had been bullied or trolled [2], and 77% of U.S. adults reported harassment [3]. Online harassment can lead to anxiety, sleeping problems, and even suicidal thoughts [3], and 66% of U.S. adult internet users strongly agree that internet trolls are detrimental to society [4]. In a survey of British Members of Parliament, all respondents reported experiences of being trolled and women reported consequent concerns about their safety [21].

Some predictors of trolling behavior have become apparent in the literature. For example, research suggests that men are more likely to troll others than women [5, 8, 22] but this gender difference is not found in all studies [7]. Moreover, some studies found that younger people are more likely to troll others [22, 23]. The literature has also showcased that personality is a promising predictor of trolling behavior. The following section will go into more detail about personality traits that predict online trolling. First, we will introduce research on the Dark Tetrad and online trolling; then, we will discuss a link between trolling and humor. Finally, we will outline why situational factors, especially social exclusion, may lead to trolling behavior.

## 1.2. The dark tetrad and online trolling

Prior research on online anti-social behavior asked, 'who trolls others?', sparking investigations of the trolls' personalities. Specifically, research has confirmed the link between trolling behavior and the Dark Tetrad traits (sadism, psychopathy, Machiavellianism, and narcissism). *Narcissism* refers to excessive self-love and a grandiose sense of self-importance [24]; *Machiavellianism* refers to the willingness to manipulate others [25]; subclinical *psychopathy* refers to fearless dominance and disinhibition [26]; and *sadism* refers to intentionally inflicting psychological/physical pain for enjoyment or power [27]. These four correlated, theoretically distinct traits share a core of callous manipulation [28]. Moreover, the Dark Tetrad facets are associated with self-reported, observer-reported, and behavioral aggression [28]. Indeed, research has confirmed the link between people scoring high on the Dark Tetrad traits and trolling behavior [5–8, 23, 29–31] which may be due to lower affective empathy in these individuals [32–34], a tendency for moral disengagement [35], and reduced behavioral inhibition anxiety [36]. Moreover, all four facets are positively associated with dominance [37] and social dominance orientation is also associated with past trolling and acceptance of trolling [38]. Another reason for the association between the Dark Tetrad and trolling behavior could be intrinsic enjoyment: Research indicates that sadism, psychopathy, Machiavellianism, and narcissism correlate positively with one's enjoyment of viewing violent stimuli [36]. Sadism is also related to experiencing greater pleasure during an aggression [39]. Moreover, a recent meta-analysis suggests that the relationship between sadism and aggressive behavior is stronger in online settings, perhaps due to anonymity [40]. Psychopathy specifically may also be related to trolling behavior because of its association with impulsivity [26, 41]. In the case of narcissism, Vize and colleagues [42] showed that antagonism primarily drives aggressive behaviors, although all narcissistic dimensions are related to aggressive behavior [43].

Table 1 showcases the correlates of trolling constructs with the Dark Tetrad in previous research. Overall, a clear pattern emerges, with higher scores on the Dark Tetrad facets being related to more self-reported trolling activities. This pattern has been confirmed for sadism in a recent meta-analysis which revealed a pooled correlation between everyday sadism and online trolling of .52 [40].

Based on this research, we aimed to confirm previous correlational findings:

H1: The Dark Tetrad is positively associated with global trolling behavior.

H1a: Machiavellianism is positively associated with global trolling behavior.

H1b: Narcissism is positively associated with global trolling behavior.

H1c: Psychopathy is positively associated with global trolling behavior.

H1d: Sadism is positively associated with global trolling behavior.

## 1.3. Humor styles and online trolling

This study also investigated the relationship between humor styles and trolling behavior. Specifically, we aimed to investigate humor as conceptualized in the humor styles questionnaire (HSQ) [45]. The theory of the HSQ assumes that humor can be adaptive or maladaptive for well-being and that people use humor to enhance the self and/or their relationships with others [45]. *Aggressive humor* refers to humor that enhances oneself while hurting others, while *self-defeating humor* is detrimental to oneself and is used to improve relationships [46]. Meanwhile, *affiliative humor* and *self-enhancing humor* are detrimental neither to oneself nor others; more specifically, affiliative humor is used to improve relationships while self-enhancing humor aims at enhancing oneself [46].

In two previous studies, active trolls stated that they engage in trolling behavior for their enjoyment [11] and instant entertainment as well as gratification [47]. This suggests that

**Table 1. Correlations between trolling assessments and the dark tetrad facets.**

| Source | Trolling construct | M | N | P | S |
|---|---|---|---|---|---|
| Buckels and colleagues (2014), study 1 | "What do you enjoy doing most on these comment sites?" | + | + | + | + |
|  | Answer option: "trolling other users" | | | | |
| Buckels and colleagues, (2014), study 2 | GAIT [a] | + | + | + | + |
| Buckels and colleagues (2019), study 1 | GAIT | + | + | + | + |
| Buckels and colleagues (2019), study 2 | iTroll | + | + | + | + |
| Craker and March (2016) | The Global Assessment of Facebook® Trolling | + | + | + | + |
| Lopes and Yu (2017) | Troll_P [b] | 0 | 0 | + | n. a. |
| Lopes and Yu (2017) | Troll_LP [c] | + | 0 | + | n. a. |
| March (2019) | GAIT-Revised | n. a. | + | + | + |
| March and colleagues (2017) | Modified GAIT | + | + | + | + |
| March and Steele (2020) [44] | GAIT-Revised | n. a. | n. a. | + | + |
| Masui (2019) | GAIT-Revised | + | + | + | + |
| Navarro-Carrillo and colleagues (2021) | GAIT | + | + | + | + |
| Nitschinsk and colleagues (2022) | Modified GAIT | n. a. | n. a. | + | + |
| Nitschinsk and colleagues (2022) | Trolling in chat room | n. a. | n. a. | 0 | + |
| Paananen & Reichl (2019) | GAIT | n. a. | n. a. | n. a. | + |
|  | Gendertrolling Measure | n. a. | n. a. | n. a. | + |
| Seigfried-Spellar & Lankford (2018) | Commenting style measure: Trolling [d] | + | + | + | + |
| Sest and March (2017) | GAIT-Revised | n. a. | n. a. | + | + |

*Notes*: M = Machiavellianism. N = narcissism. P = psychopathy. S = sadism [a] Global Assessment of Internet Trolling; [b] Agreement score to trolling comments towards the popular Facebook profile; [c] Agreement score to trolling comments towards the less popular Facebook profile; [d] developed by authors, no further information available; n. a. = not assessed

internet users' humor styles may also affect whether they troll others online. Indeed, one study [48] recently showed that trolling behavior is associated with aggressive humor as well as katagelasticism (i.e., the joy of laughing at others). Moreover, self-enhancing and self-defeating humor were both related to trolling behavior in that study. Some authors suggest that katagelasticism may be a cause of trolling behavior [49]. Additionally, aggressive humor is positively associated with the readiness to be verbally aggressive [50] which, we expect, may be expressed in trolling. In fact, sarcasm and mockery (aggressive forms of humor [51]) can be tactics trolls use to disrupt discussions [20, 52].

Previous research has also shown that humor styles are associated with the Dark Tetrad. The four dark personality traits are linked with inadequate humor [32], e.g., schadenfreude in social, academic, and mourning contexts [35, 53]. Moreover, humor research indicates that the Dark Tetrad facets are linked to how people utilize and enjoy humor [54], e.g., higher Machiavellianism and subclinical psychopathy scores are associated with aggressive humor [55, 56]. Specifically, psychopathy appears to have a stronger connection with aggressive humor and katagelasticism than Machiavellianism and narcissism [54]. Sadism also uniquely explains variance in katagelasticism beyond the Dark Triad [54]. Importantly, katagelasticism not only involves enjoying laughing at others but also actively seeking out situations where one can ridicule others [54]. We believe that internet trolling may be an expression of katagelasticism in people who score highly on the Dark Tetrad facets. In sum, it is likely that trolling behavior is associated with more aggressive humor and this link may exist because people who score highly on the Dark Tetrad use humor differently than people who score lower on the Dark Tetrad. Based on the prior research, we had one further hypothesis and one research question:

H2: Aggressive humor is positively associated with trolling behavior.

RQ: How do affiliative, self-defeating, and self-enhancing humor relate to trolling behavior?

## 1.4. Social exclusion and online trolling

While findings of the Dark Tetrad traits and internet trolling are relatively consistent across studies, it is important to keep in mind that other variables may be of importance as well: Naturally, behaviors can be influenced by individual (e.g., a sadistic personality) and situational (e.g., being exposed to aggressive behaviors from others) factors and this should also be the case for trolling behavior. For example, in a study in which trolling behavior was assessed by rating comments the participants made, the authors found that negative mood (and exposure to troll comments) triggers trolling behavior [9]. Other studies found that anonymity in chat rooms led to more troll comments than a chat room condition where participants were identifiable [12]. Additionally, research suggests that boredom in life [57] and loneliness [58] predict trolling. These studies highlight the importance of situational factors (here negative mood, exposure to trolls, anonymity, boredom, and loneliness) in the context of internet trolls.

To the authors' best knowledge, most studies on trolling behavior neglect the investigation of external variables; however, one experiment showed that participants who were socially excluded through a mobile phone text messaging set-up with two other people wrote more provocative messages afterward [10]. Thus, one of the situational factors that might increase trolling behavior could be social exclusion: Revenge can be a motivation for trolling behavior [11] and aggressive responses to rejection can occur, for example, to regain control [59–61]. This aligns with the meta-analytic finding that the relationship between narcissism and aggression is stronger after provocation [43]. One recent study [62] showed that social exclusion compared to social inclusion leads to significantly higher cyberaggression in narcissistic individuals. Moreover, cyberaggression has been found to relate to loneliness and being less socially accepted [63]. Additionally, in the above-mentioned study, socially excluded participants not only wrote more aggressive text messages but also reported worse mood [10], which has predicted writing trolling comments in other research [9]. Importantly, scoring highly on Dark Tetrad traits is associated with difficulties in emotion regulation [32, 64, 65]; hence, we believe that exclusion experiences may be difficult to process for trolls which then may lead to trolling behavior.

Hence, based on the prior research described above [10, 11, 62], we suggest that it might also be possible that feeling excluded motivates people to troll other internet users to avenge themselves. With quick access to social media and the potential of social exclusion occurring online, trolling posts/comments might be an easy way for people who just experienced social exclusion to regain their perceived control [59–61]. Consequently, we intend to add to the base of knowledge of 'who trolls?' by also asking 'when?'. To that end, not only did we investigate global trolling behavior, i.e., a person's general trolling behavior, but also immediate trolling motivation after an exclusion experience. We hypothesized:

H3: Participants who are socially excluded show increased immediate trolling motivation compared to people who are socially included.

As shown in Table 1, the Dark Tetrad facets correlate positively with trolling behavior. However, a new pattern emerges when looking at multiple regression results rather than biserial correlates: In prior research, only some of the Dark Tetrad facets explained a significant amount of variance in trolling behavior when looking at partial correlations or multiple regression [6, 8, 22, 23, 58]. Specifically, in some studies [23, 66], only sadism and psychopathic tendencies explained a significant amount of variance in trolling behavior when all facets of the

Dark Tetrad were included in a multiple regression analysis. In contrast, earlier work revealed sadism and Machiavellianism as significant positive predictors of trolling enjoyment, while psychopathy was unrelated and narcissism showed a negative association [5]. Other research shows significant positive effects of Dark Tetrad facets except for narcissism [58]. Overall, sadistic tendencies appear to have a greater impact on trolling behavior than the other facets [32]. Hence, while all facets of the Dark Tetrad appear to correlate with trolling behavior, in multiple regression analyses, psychopathic personality traits as well as sadism appear to be more consistent predictors of internet trolling than Machiavellianism and narcissism. This highlights the need to investigate the differing roles of the Dark Tetrad facets in more detail.

H4: Social exclusion and Machiavellianism, narcissism, psychopathy, and/or sadism can predict immediate trolling motivation.

The present study aims at expanding prior findings by (a) confirming the previously established predictive role of Dark Tetrad traits, (b) investigating the role of humor styles as predictors of global trolling behavior, and (c) testing the impact of a situational variable, namely social exclusion, on immediate trolling motivation. Based on these aims, we conducted an experiment using the Cyberball paradigm and tested the effects of inclusion and exclusion on the participants' immediate trolling motivation.

## 2. Method

### 2.1. Participants

This study includes 1,026 participants ($M_{age}$ = 26.46 ($SD_{age}$ = 5.88); 77.2% female) recruited from four German universities and a popular science website for psychology (https://www.psychologie-heute.de/aktuelles/studienteilnahme.html). Our sample size surpasses the necessary sample size of 260 participants required to detect an effect of .18 (at the time of the pre-registration smallest reported effect of the correlations between trolling behavior and the Dark Tetrad [22]) for an alpha of .05 and a power of .90.

### 2.2. Materials

Demographic questions assessed participants' age, gender, sexuality, nationality, and favorite social media platform. It was also assessed if participants had a fake account and, if they had fake accounts, on which platforms and for what purposes.

**Global and immediate trolling behavior.** To assess global trolling behavior, we used the revised Global Assessment of Internet Trolling (GAIT-Revised) [8]. This is an 8-item self-report measure that assesses trolling behavior online (e.g., "Although some people think my posts/comments are offensive, I think they are funny."). Items are rated on a 5-point scale from 1 (*strongly disagree*) to 5 (*strongly agree*), Cronbach's alpha = .60. As no validated German versions exist for the GAIT-Revised yet, the authors translated this measure and discussed the item formulations. The translation process involved several revisions whereupon each revision aimed to maximize the accuracy of the translation while simultaneously maximizing the naturalness of the German formulation. All people involved in this process were fluent in English and German and familiar with the concept of internet trolling. Next to the GAIT-Revised as a global measure for trolling behavior, this study also assessed immediate motivation to troll others (Immediate Assessment of Internet Trolling, IAIT). The authors created the IAIT measure by reformulating the items to have them address the present moment (e.g., "Just now, I want to share posts/comments that I think are funny, although some people might think they are offensive."). This process also involved several steps during which the item formulations were clarified and improved. For this adaptation, as with the GAIT, the authors critically investigated the item formulations for understandability and face validity. Cronbach's

alpha for all items was = .54, by excluding item 6 ("I prefer not to cause controversy or stir up trouble right now"), we achieved a Cronbach's alpha of .70 for the IAIT.

**Dark tetrad.** The Short Dark Triad (SD3) [67, 68], a 27-item scale, was used to measure Machiavellianism (e.g., "I like to use clever manipulation to get my way.", Cronbach's alpha = .76), narcissism (e.g., "Many group activities tend to be dull without me.", Cronbach's alpha = .73), and subclinical psychopathy (e.g., "People who mess with me always regret it.", Cronbach's alpha = .71). We used a published, validated German translation of the SD3 [68]. Items are rated on a 5-point scale from 1 (*strongly disagree*) to 5 (*strongly agree*). To assess sadism, the Comprehensive Assessment of Sadistic Tendencies (CAST) [69] was used. The CAST is an 18-item self-report measure that assesses sadistic personality. The CAST can be divided into three dimensions: direct verbal sadism (e.g., "I was purposely mean to some people in high school."), direct physical sadism (e.g., "I enjoy physically hurting people."), and vicarious sadism (e.g., "In video games, I like the realistic blood spurts."). Items are rated on a 5-point scale from 1 (*strongly disagree*) to 5 (*strongly agree*). As no validated German versions exist for the CAST yet, the authors translated this measure and discussed the item formulations in several steps to match them as closely as possible to the original English and maximize the naturalness of the German translation. Cronbach's alpha for the CAST = .75.

**Humor styles.** To assess humor styles, we used the Humor Styles Questionnaire (HSQ) [46, 47], a 32-item self-report measure, that comprises of four different humor styles: self-enhancing (e.g., "If I am feeling depressed, I can usually cheer myself up with humor.", Cronbach's alpha = .84.), affiliative (e.g., "I laugh and joke a lot with my friends.", Cronbach's alpha = .81.), aggressive (e.g., "If I don't like someone, I often use humor or teasing to put them down.", Cronbach's alpha = .70.), and self-defeating (e.g., "I often try to make people like or accept me more by saying something funny about my own weaknesses, blunders, or faults.", Cronbach's alpha = .75.). Items are rated on a 7-point scale from 1 (*totally disagree*) to 7 (*totally agree*). While no official validation study for the German HSQ exists, there is support for the factorial validity of the German translation we used [47].

**Experimental manipulation.** To manipulate social exclusion, we used the Cyberball paradigm [70]. Cyberball is an experimental manipulation that leads participants to believe that they are playing an online ball-tossing game with two other study participants. In reality, the behavior of the other players is programmed. In this study, participants were either excluded or included in the ball-tossing game. In the inclusion condition, participants got the ball ten times out of 30 tosses. In the exclusion condition, participants got the ball only one time. Using 30 throws is common practice in social exclusion studies [71].

## 2.3. Procedure

This study was preregistered before data collection (https://osf.io/qsfe5) and adheres to Section 15 of the Professional Code of Conduct for Physicians in Bavaria; hence no vote by the Ethics Committee was necessary. All participants provided online informed consent in accordance with the Declaration of Helsinki. Before the study, participants received information about the context of the research, that all information was anonymous and that they were free to discontinue the study at any point in time. To begin the study, participants had to click on a field that read "I agree with these conditions and want to proceed.". Minors were not allowed to participate in this research. After giving informed consent, participants answered the GAIT-Revised, SD3, CAST, and HSQ and demographic items. Then, participants were randomly allocated to either a social exclusion or inclusion condition using the Cyberball paradigm. After the Cyberball manipulation, participants were asked to rate their immediate motivation to troll others (IAIT). Following this, study subjects received an explanation of the study's actual goal.

Throughout the survey, we placed three attention-check items. If participants gave wrong answers to these items, the experiment immediately ended and brought the subjects to an explanation page.

## 2.4. Analysis

Before the analysis, people were excluded when they indicated that they had not taken the questionnaire seriously (participants were asked directly if they had taken the questionnaire seriously), were already familiar with Cyberball or were unable to see the ball-tossing program, and when they were below the age of 18. Additionally, because some answers were randomly missing due to the questionnaire software, we excluded cases listwise for missing data. Finally, because only one person indicated their gender as "other than male or female," this participant was also excluded from further analysis. This was necessary since no reliable inferences can be drawn from a sample of only one person.

In our pre-registration, we planned to use means of the measures for the analyses. However, after investigating the item loadings of our measures using confirmatory factor analysis (see S1 Appendix), we were concerned about the validity of the scales when using all items. Consequently, we conducted all analyses twice, once using the means with all items as preregistered and once using means that only included items that had a standardized loading on its factor of at least .40. This also serves as a robustness check of our analyses. We report our analyses with the traditional means in the Results section below. We report the analysis using means without low loading items in the S1 Appendix.

We preregistered an ordinary least square (OLS) multiple regression analysis to test H4 which aimed to predict immediate trolling motivation using social exclusion (yes/no) and the Dark Tetrad (Machiavellianism, narcissism, psychopathy, and sadism). Additionally, age and gender were added as control variables, as those variables have been shown to explain trolling behavior in the past [22]. Our regression assumption checks showed some violations of the homoscedasticity and normal distribution of errors assumption. Hence, we decided to deviate from our pre-registration and conducted a quantile regression analysis, which allows for residuals to have different variances [72] and does not assume parametric distributional form (here normal) of the errors [73]. Consequently, a quantile regression analysis should be better suited for our data. Moreover, prior research has compared simple linear regression with quantile regression for personality trait data and concluded that quantile regression can showcase more nuanced and heterogeneous effects [74, 75].

Unlike OLS regression, quantile regression relies on quantiles of the outcome variable which results in several coefficients for a single covariate [76]. These coefficients are interpreted based on their respective quantile of the outcome variable [76]. For example, while OLS regression may indicate an average effect of a woman's partner's meanness on relationship satisfaction, quantile regression can show that for the 15th quantile of relationship satisfaction (i.e., the least satisfied women), the partner's meanness can reduce relationship satisfaction by 0.61 points, while partner's meanness only reduces relationship satisfaction by 0.14 points for the 85th quantile (i.e., the most satisfied women) [77].

Finally, we decided to conduct a dominance analysis that was not pre-registered. We based our decision on criticism of the use of multivariate statistics in Dark Tetrad research [78]. Dominance analysis can provide indicators of relative predictor importance [79, 80]. Dominance analysis calls one predictor more important than another "if it would be chosen over its competitor in all possible subset models where only one predictor of the pair is to be entered" (p. 134) [79]. The dominance analysis approach should provide more definite answers to the question which of the Dark Tetrad facets are the most relevant to predict trolling than multiple

regression. The approach has been used previously to answer similar questions, for example, to estimate which narcissism subcomponent is most important for predicting aggressive behavior [42]. We consider the dominance analysis here as an additional robustness check. We tested the importance of all regression predictor variables of our H4 model. The analysis can be found in the S1 Appendix.

We used the R-packages "lavaan" [81] to conduct our confirmatory factor analyses, "quantreg" [82] to conduct our quantile regression analysis, and "dominanceanalysis" [83] to conduct our dominance analysis (R version 4.1.2). For the remaining analyses, we used SPSS (version 26). For our quantile regression, our solution of the means without low loading items resulted in a non-unique solution. Restricting the tau range from 5:95 to 10:90 resulted in a unique solution and is reported in the S1 Appendix.

## 3. Results

### 3.1. Sample descriptives

Overall, 1,026 people participated in this study; $M_{age}$ = 26.46 ($SD_{age}$ = 5.88); 77.2% were female and 22.8% were male. Table 2 describes the study's sample concerning age and our continuous constructs.

### 3.2. Global trolling behavior and personality traits

**3.2.1. Global trolling behavior and the dark tetrad.** To test our first hypothesis (H1: The Dark Tetrad is positively associated with global trolling behavior) and the respective sub-hypotheses, we looked at the correlations between Dark Tetrad personality scores and global trolling behavior; see Table 3. Our results indicate that each of the Dark Tetrad personality facets correlates positively and significantly with global trolling behavior. Thus, these findings support H1. Please note that we pre-registered tests for H1 and H2 one-sided but tested two-sided for significance.

**3.2.2. Global trolling behavior and humor styles.** To test our second hypothesis (H2: Aggressive humor is positively associated with trolling behavior) and our research question (How do affiliative, self-defeating, and self-enhancing humor relate to trolling behavior?), we checked the correlations between global trolling behavior and the four humor styles; see Table 3. As predicted, higher aggressive humor was significantly associated with more global

**Table 2. Sample descriptives.**

|  | *M* | *SD* | Min | Max | Potential range |
|---|---|---|---|---|---|
| Age | 26.46 | 5.88 | 18 | 77 | 18–99 |
| Global trolling | 1.45 | 0.43 | 1 | 4 | 1–5 |
| Immediate trolling motivation | 1.18 | 0.33 | 1 | 3.71 | 1–5 |
| Machiavellianism | 2.94 | 0.63 | 1 | 4.89 | 1–5 |
| Narcissism | 2.69 | 0.61 | 1.11 | 4.78 | 1–5 |
| Psychopathy | 1.97 | 0.58 | 1 | 4.33 | 1–5 |
| Sadism | 1.58 | 0.47 | 1 | 3.94 | 1–5 |
| Aggressive humor | 3.17 | 0.95 | 1 | 6.25 | 1–7 |
| Affiliative humor | 5.81 | 0.83 | 1.88 | 7 | 1–7 |
| Self-enhancing humor | 4.64 | 1.05 | 1 | 7 | 1–7 |
| Self-defeating humor | 3.27 | 1.12 | 1 | 6.38 | 1–7 |

*Note.* n = 1,026

**Table 3. Correlation matrix for trolling, the dark tetrad of personality, and humor styles.**

| | 2. | 3. | 4. | 5. | 6. | 7. | 8. | 9. | 10. |
|---|---|---|---|---|---|---|---|---|---|
| 1. Global trolling | .49 ** | .31** | .34** | .49** | .48** | .36** | .01 | .04 | .12** |
| 2. Immediate trolling motivation | 1 | .24** | .23** | .42** | .42** | .31** | .01 | -.05 | .14** |
| 3. Machiavellianism | | 1 | .35** | .56** | .44** | .37** | -.06 | .03 | .16** |
| 4. Narcissism | | | 1 | .46** | .38** | .32** | .17** | .26** | .05 |
| 5. Psychopathy | | | | 1 | .66** | .53** | -.06 | .03 | .21** |
| 6. Sadism | | | | | 1 | .58** | -.03 | .05 | .16** |
| 7. Aggressive humor | | | | | | 1 | .001 | .12** | .24** |
| 8. Self-enhancing humor | | | | | | | 1 | .37** | .06 |
| 9. Affiliative humor | | | | | | | | 1 | .10** |
| 10. Self-defeating humor | | | | | | | | | 1 |

*Note.* $n = 1.026$

** $p < .01$ (two-tailed)

trolling, $r(1024) = .36$, $p < .01$. Next to aggressive humor, self-defeating humor was also significantly associated with trolling behavior, $r(1024) = .12$, $p < .01$, while affiliative humor and self-enhancing humor showed no significant relationship with trolling. Thus, these findings support H2 and answer our research question.

### 3.3. Immediate trolling motivation and social exclusion

To test our third hypothesis (H3: Participants who are socially excluded show increased immediate trolling motivation compared to people who are socially included), we conducted a *t*-test with exclusion (yes/no) as the independent and immediate trolling motivation as the dependent variable. Our result suggests that the experience of exclusion did not significantly impact participants' immediate trolling motivation, $t(1024) = 0.91$, $p = .37$, CI = [-.02; .06]. Thus, findings from this analysis did not support H3.

### 3.4. Predicting immediate trolling motivation

The quantile regression results to test our fourth hypothesis (H4: Social exclusion and Machiavellianism, narcissism, psychopathy, and/or sadism can predict immediate trolling motivation) are graphically presented in Fig 1. For example, the third graph in the first row of Fig 1 shows how Machiavellianism is predictive of immediate trolling motivation. The red horizontal line represents the ordinary least square (OLS) coefficient for Machiavellianism, while the x-axis represents the quantiles (at 0.2, 0.4, 0.6, and 0.8) of immediate trolling motivation. Furthermore, the black broken line indicates the coefficients at the respective quantiles, here ranging from the 0.5 to the 0.95 quantile in 0.10 steps. Machiavellianism does not seem to predict immediate trolling motivation for the 0.2 quantile of immediate trolling motivation, while for the 0.6 quantile, Machiavellianism appears to have a positive (albeit non-significant, see Table 4) effect on immediate trolling motivation.

Table 4 provides the quantile regression coefficients and their significance as well as OLS regression results. A comparison between OLS and quantile coefficients reveals some significant differences: For example, the OLS regression coefficient for sadism indicates that for every 1-point increase on the sadism scale, immediate trolling motivation also increases by 0.20. In contrast, the quantile analysis shows that there is no significant relationship between sadism and immediate trolling motivation for the quantiles 0.05 to 0.35.

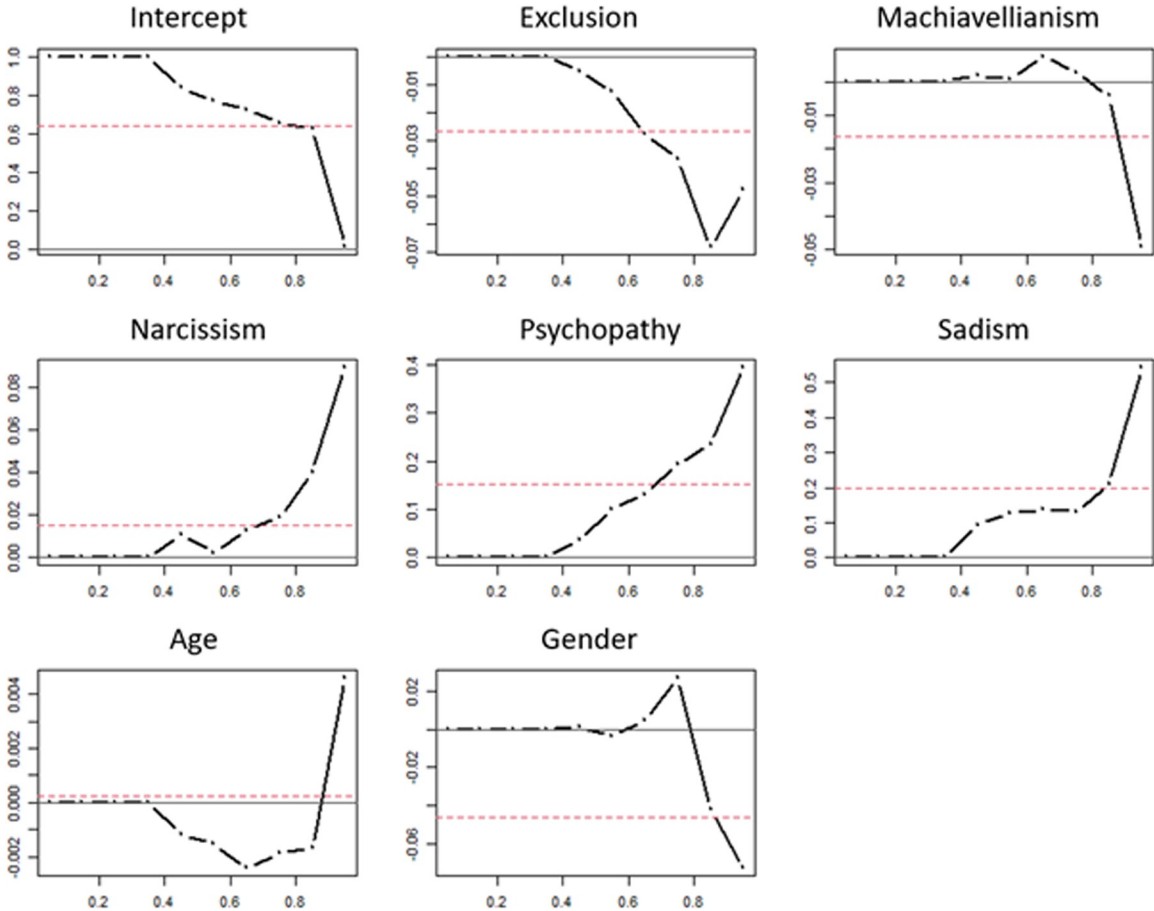

**Fig 1. Graphs of the quantile regression coefficients for all independent variables.** *Note.* Simple linear regression coefficients (red line) and quantile regressions for exclusion, Machiavellianism, narcissism, psychopathy, sadism, age, and gender (male with female as the comparison group) for the dependent variable immediate trolling motivation. The x-axis represents the quantiles for immediate trolling motivation while the y-axis represents the unstandardized coefficients of the respective independent variable.

Unlike the OLS coefficient for social exclusion, the exclusion coefficients for the 0.75 and the 0.85 quantiles are significant and–in contrast to our predictions–negative. This indicates that experiencing social exclusion may reduce immediate trolling motivation for people who are part of the upper immediate trolling motivation quantiles.

Thus, quantile regression allowed us to provide a more nuanced view of the relationship between our independent variables and trolling motivation. Overall, we find no significant effects of exclusion experience, Machiavellianism, and narcissism on immediate trolling motivation. Meanwhile, for the 0.45 quantile and higher quantiles of immediate trolling motivation, psychopathy and sadism appear to become increasingly relevant.

### 3.5. Robustness of findings

Due to concerns about construct validity, we conducted CFAs and consequently reran our analysis using means that excluded low loading items (see S1 Appendix). Overall, our robustness analysis replicates the correlational findings and the *t*-test finding. However, the quantile regressions differ to some degree. After excluding low-loading items, only sadism predicted immediate trolling motivation for higher quantiles of immediate trolling motivation. In other

**Table 4. Quantile regression coefficients of exclusion, machiavellianism, narcissism, psychopathy, sadism, age, and gender for the dependent variable "immediate trolling motivation".**

| Quantiles | Quantile regression coefficients | | | | | | | |
|---|---|---|---|---|---|---|---|---|
| | Intercept | Exclusion | Machiavellianism | Narcissism | Psychopathy | Sadism | Age | Male |
| 0.05 | 1*** | 0 | 0 | 0 | 0 | 0 | 0 | 0 |
| 0.15 | 1*** | 0 | 0 | 0 | 0 | 0 | 0 | 0 |
| 0.25 | 1*** | 0 | 0 | 0 | 0 | 0 | 0 | 0 |
| 0.35 | 1*** | 0 | 0 | 0 | 0 | 0 | 0 | 0 |
| 0.45 | .836*** | -.01 | .002 | .01 | .04* | .09** | -.001** | .001 |
| 0.55 | .77*** | -.01 | .001 | .002 | .10*** | .13*** | -.002* | -.003 |
| 0.65 | .72*** | -.03 | .01 | .01 | .13*** | .14** | -.002* | .005 |
| 0.75 | .66*** | -.04* | .003 | .02 | .19*** | .13ᵗ | -.002 | .03 |
| 0.85 | .63*** | -.07* | -.004 | .04 | .23*** | .21* | -.002 | -.04 |
| 0.95 | .01 | -.05 | -.05 | .09 | .39** | .54** | .005 | -.07 |
| | OLS regression coefficients (SE) [BCa-95% CI] | | | | | | | |
| | Intercept | Exclusion | Machiavellianism | Narcissism | Psychopathy | Sadism | Age | Male |
| | .64 (.07) [.46; .83] | -.03 (.02) [-.06; .01] | -.02 (.02) [-.06; .02] | .02 (.02) [-.02; .05] | .15 (.02) [.10; .21] | .20 (.03) [.11; .28] | < .001 (.002) [-.003; .004] | -.05 (.03) [-.12; .03] |

*Note.* ᵗ $p < .10$

\* $p < .05$

\*\* $p < .01$

\*\*\* $p < .001$. OLS = ordinary least square.

words, neither Machiavellianism nor narcissism *or* psychopathy uniquely predicted immediate trolling motivation when we controlled for low loading items.

This finding is mirrored by our additional dominance analyses (see S1 Appendix): When using the traditional means for our dominance analysis, both sadism and psychopathy were the most important predictors for immediate trolling motivation. However, when we used means without low loading items, sadism became the most important predictor and dominated psychopathy with its predictive power.

## 4. Discussion

In this pre-registered study, we investigated trolling behavior and its association with the Dark Tetrad and humor styles. We found support for our first hypothesis (H1: The Dark Tetrad is positively associated with global trolling behavior): Machiavellianism, narcissism, psychopathy, and sadism showed significant positive correlations with global trolling behavior. Moreover, we found support for our second hypothesis (H2: Aggressive humor is positively associated with trolling behavior) and also observed a positive correlation between global trolling behavior and self-defeating humor. In contrast to our expectations (H3: Participants who are socially excluded show increased immediate trolling motivation compared to people who are socially included), we found no effect of social exclusion experience on immediate trolling motivation. Consequently, we could also only partly accept our fourth hypothesis (H4: Social exclusion and Machiavellianism, narcissism, psychopathy, and/or sadism can predict immediate trolling motivation), as psychopathy and sadism but not Machiavellianism nor narcissism were significant predictors of immediate trolling motivation for the higher quantiles of immediate trolling motivation. Moreover, the findings concerning psychopathy should be considered with caution since sadism was the only significant predictor in our robustness quantile regression. Though the effect of social exclusion was generally non-significant, we found

significant and negative effects for the 0.75 and 0.85 quantiles of immediate trolling motivation. We will now discuss these findings in the context of current research.

First, our results validate the correlational association between trolling and dark personality traits in the German-speaking context. However, our correlations were only in the small to moderate range, which somewhat contrasts with prior research in which trolling-personality correlations of up to $r = .71$ were sometimes reported [23].

Second, and in contrast to our predictions, experiencing social exclusion did not (always) lead to stronger immediate motivation to troll others. Indeed, social exclusion was a significant predictor of immediate trolling motivation for the 0.75 and the 0.85 immediate trolling motivation quantile, with being excluded appearing to *reduce* motivation to troll others. In our robustness quantile regression, social exclusion did not predict immediate trolling motivation for any quantile of immediate trolling motivation. This result stands out because exclusion has been shown to lower one's mood [59], and a prior study [9] suggests that bad mood can contribute to trolling behavior. However, it should be noted that the assessment of trolling behavior in the present study differed from the one used by Cheng and colleagues (2017) [9]. The present study assessed participants' immediate motivation to troll by relying on a self-report measure. This was done due to technical restrictions in our study design and the survey platform. In contrast, Cheng et al. (2017) asked participants to interact in a comment section under a short news article, and comments were then rated as troll comments (yes/no) by two independent experts. Thus, their results might have occurred because participants had an immediate chance to act on impulses to troll others, whereas subjects in the present study were asked about their intentions. As such, the difference in results may be due to an intention-behavior gap. We assessed immediate trolling motivation using a self-report measure rather than assessing actual trolling behavior unobtrusively (e.g., by letting people write comments and rating trolling content). Using self-reports rather than unobtrusive approaches has been shown to result in a bias towards socially desirable responses [84, 85]. However, it might also be the case that Cyberball was not a sufficient manipulation or that ostracism does not always lead to aggressive or revengeful behavior. A meta-analysis [59] specifies that rejection does not necessarily have to result in aggression: Following rejection, people act antisocially to satisfy a need for control that could otherwise not be achieved. In our context, participants were excluded by strangers whom the participants would never meet again after a five-minute game. It might be the case that their perceived control was not reduced enough to react in an antisocial way.

Another explanation for our generally non-significant results for social exclusion, as well as the two observed negative effects of social exclusion on trolling motivation, may be due to one exclusion experience not leading to immediate aggression: Short-term social exclusion generally appears to lead to behaviors meant to ameliorate the situation so long as control can be regained [86, 87]. In this context, the two negative effects of social exclusion on immediate trolling motivation make sense, as they could be understood as a way to regain affiliative opportunities. In comparison, long-term social exclusion may result in the temporarily aggressive self becoming a person's actual self [88]. Hence, we might hypothesize that long-term rather than short-term social exclusion may lead to trolling behavior.

Finally, it is important to note that our assessment of trolling motivation was not person-specific; in other words, we did not ask if participants wanted to troll the people who had just excluded them. Cook and colleagues (2018) [11] found that revenge is a reason to troll others for self-confessed trolls and that this is a response to others behaving 'stupidly' (p. 3332) or being trolled themselves. Based on these points, it might prove valuable to re-examine the effect of exclusion on trolling behavior in a more realistic context where people have the chance to target users who ostracized them.

Third, this study showed that psychopathy and sadism predict immediate trolling motivation, whereas other dark personality facets did not. Moreover, in our robustness analysis, only sadism predicted immediate trolling motivation. These findings are not completely surprising, as other multiple regression analyses also do not find that each of the Dark Tetrad facets explains a significant amount of variation in trolling behavior. For example, in one study [5], neither Machiavellianism nor narcissism significantly predicted trolling, and the same result was found by Craker and March (2016) [22] for trolling behavior on Facebook. Thus, it appears that psychopathy and sadism are significant predictors of trolling when all facets of the Dark Tetrad are taken into account. In contrast, Machiavellianism and narcissism probably do not explain any variance in immediate trolling motivation when psychopathy and sadism are controlled for. A recent review indicates (a) that sadism generally motivates trolling more than the remaining Dark Tetrad facets and (b) that the association of sadism with psychopathy is stronger compared to the relationships with narcissism or Machiavellianism [32]. The same review also suggests that different aggressive behaviors show stronger associations with sadism and psychopathy but not necessarily with narcissism [32]. The finding that sadism is a better predictor of trolling than the Dark Triad facets [32] mirrors our robustness quantile regression where, after low loading items were excluded, only sadism predicted trolling motivation.

These patterns may be due to the stronger association between sadism and psychopathy with aggressive behaviors, but it may also be due to the often one-dimensional assessment of the Dark Triad facets [78]. The Dark Triad research has been criticized because measures often neglect the multidimensionality of the Dark Triad facets [78]. Investigating subcomponents of the Dark Tetrad facets might have provided more nuanced insights into the relationships between personality and trolling behavior. For example, Vize and colleagues [42] investigated different facets of narcissism and found that grandiose narcissism was more important in explaining proactive aggression whereas vulnerable narcissism was more important for reactive aggression. In the context of our experiment which manipulated social exclusion, a differentiation between grandiose and vulnerable narcissism might have aided our understanding.

The concept of the Dark Tetrad has received further criticism: Some researchers suggest that narcissism and Machiavellianism are features of psychopathy and that, thus, the Dark Triad does not explain variance beyond psychopathy [89]. Finally, humor research has indicated that psychopathy outperforms the other facets of the Dark Tetrad in explaining aggressive humor and katagelasticism [54] which may instigate trolling behavior [49]. Sadism also uniquely predicts katagelasticism although to a lesser degree than psychopathy [54]. In contrast, narcissism is associated with lighter forms of humor that enable relationship-building while Machiavellianism is strongly associated with the use of irony and the fear of being laughed at [54].

As such, several potential reasons for our findings arise: (a) Psychopathy and sadism are stronger predictors of aggressive behavior, (b) we did not assess the multidimensionality of narcissism, Machiavellianism, and psychopathy, (c) effects of narcissism and Machiavellianism may already be explained by including psychopathy due to an overlap in definitions, and (d) psychopathy and sadism appear to have stronger associations with katagelasticism than narcissism and Machiavellianism. Finally, sadism may outperform psychopathy in our robustness analysis, since people with high sadistic tendencies feel greater aggressive pleasure which may motivate people to behave aggressively [39], here: motivation to troll others. This intrinsic enjoyment of inflicting pain is not a crucial component of psychopathy [90].

Fourth, our quantile regression showed that even psychopathy and sadism have no explanatory power for the lower quantiles of immediate trolling motivation. However, for higher quantiles of immediate trolling motivation (i.e., people who were in the higher percentiles of

immediate trolling motivation), psychopathy and sadism become stronger predictors. This finding highlights the importance of quantile regression in personality research and suggests that even psychopathy and sadism may not be suitable predictors for no or minimal trolling behavior.

Lastly, this study found significant associations between trolling and aggressive as well as self-defeating humor, confirming the recent findings of Navarro-Carrillo and colleagues (2021). The positive relationship between aggressive humor and trolling behavior is in line with prior research showing links between aggressive humor and at least some dark personality traits [55, 56]. Our finding also fits with Hardaker's (2010) [1] definition of trolls, which states that they create conflict for their own amusement.

The association between trolling and self-defeating humor might appear less intuitive. Despite this surface-level contradiction, there are potential explanations for this finding: Aggressive humor has been shown to correlate with self-defeating humor. In Martin and colleagues' (2003) [46] study, this relationship was significant for men and women. Moreover, despite a troll's egocentric tendencies, they may still lack self-confidence [48].

## 4.1. Limitations and further research

This study has some limitations: Firstly, since there is no instrument to measure immediate trolling motivation, we used an adapted version of the GAIT [8]. Because no official and validated German scales of the GAIT and CAST existed, we used our translation of the scales. While we aimed to capture the original meaning of the items and the concepts while maximizing the naturalness of the German formulations, this remains a limitation of the present study. Due to concerns about the scale validity, we ran robustness analyses where we excluded low loading items. This led to partially differing results for our quantile regressions and our dominance analyses. Consequently, we urge researchers to translate and validate scales to allow for more rigorous research across different cultures.

Moreover, assessing immediate trolling behaviors (observable) rather than motivation (self-report measure) could prove more fruitful in future online trolling research, since answering a questionnaire may result in socially desirable responses [84, 85] which could make it difficult to find true effects. Additionally, being asked about trolling motivation after an experimental manipulation might hint at the study's purpose for participants. To investigate online trolling behavior further, more immersive and ecologically valid assessments (as done by Cheng and colleagues (2017) [9]) should be applied. This could even be done through field experiments in online multiplayer games.

One limitation of this study is that we did not include a manipulation check for the social exclusion manipulation. We did not include a manipulation check to avoid making the purpose of the present study apparent to participants and because a meta-analysis of 120 studies found large (d > 1.4) and generalizable effects for the Cyberball manipulation [71].

The present study only investigated malicious, serious trolling. However, another, humorous form of trolling behavior also exists [20], and to the best of our knowledge, this type of trolling has not received much attention in prior personality and humor research. Future studies may want to investigate potential differences in personality facets and humor styles between serious and humorous trolling and whether the same people use both forms of online interaction depending on different situational circumstances.

While our study has some limitations, it also provides some new avenues for further research. We suggest that future trolling research should consider criticisms of the Dark Tetrad [78, 89] and propose two strategies in future study designs. First, we suggest that researchers take the multidimensionality of the Dark Tetrad facets into account and select measures which

address the complexity of the dark personality traits. This has already been done in part by Paananen and Reichl [30] when they used verbal, physical, and vicarious sadism, and by March [29] who differentiated between direct and vicarious sadism as well as between primary and secondary psychopathy. We believe that these approaches can help to provide a more nuanced perspective on trolling behavior. Second, we suggest dominance analysis [79, 80] to compare how well components and subcomponents of the Dark Tetrad explain trolling behavior. This analysis has demonstrated, for example, that the narcissism factor interpersonal antagonism explains more variance in aggressive and antisocial behaviors than the facets of extraversion and neuroticism do [42].

Concerning humor styles, an interesting finding of our study is the positive correlation between global trolling behavior and self-defeating humor. It appears that trolls (online or offline) do not only target others but also themselves, using humor. A direction to approach this further may be to include the role of self-esteem as a predictor of trolling behaviors. Moreover, the research on trolling might want to investigate antecedents of perceived funniness [91] to better understand how and why trolls attempt their aggressive versions of comedy. This may also prove fruitful in the context of investigating different kinds of trolling [20].

Finally, we suggest moving beyond correlational studies when investigating trolling research specifically and antisocial social media behavior generally. Correlations between personality facets and trolling behavior do not explain why people choose to troll others *when* they do. We suggest a thorough examination of (potential) trolling behavior under different situational circumstances to enhance our understanding of internet trolls. This could also help platform providers to create more harmonious communities, e.g., by answering the question of whether certain functions (downvotes, likes, having featured comments, etc.) encourage people to engage in trolling behavior.

## 5. Conclusion

The present study confirms the correlational association between the Dark Tetrad of personality traits but shows that Machiavellianism and narcissism do not predict immediate trolling motivation when we control for participants' psychopathy and sadism. Moreover, even psychopathy and sadism are only significant predictors for higher quantiles of immediate trolling motivation in our main analysis. In our robustness analysis, only sadism predicted higher quantiles of immediate trolling motivation. Sadism was also notably more important than psychopathy in our robustness dominance analysis. As such, this study highlights that not all facets of the Dark Tetrad are equally predictive of trolling and that the relationship between the dark personality dimensions and trolling behavior is more nuanced than previously assumed. This also emphasizes the need for quantile regression in personality research.

For some higher quantiles of immediate trolling motivation, we found that a social exclusion experience reduced the motivation to troll others. This highlights the importance of more experimental studies to enhance our understanding of online trolls.

## Supporting information

**S1 Appendix.**
(DOCX)

## Author Contributions

**Conceptualization:** Sara Alida Volkmer, Eva Lermer.

**Data curation:** Sara Alida Volkmer.

**Formal analysis:** Sara Alida Volkmer.

**Investigation:** Sara Alida Volkmer.

**Methodology:** Sara Alida Volkmer, Eva Lermer.

**Project administration:** Eva Lermer.

**Resources:** Susanne Gaube, Eva Lermer.

**Supervision:** Susanne Gaube, Eva Lermer.

**Visualization:** Sara Alida Volkmer.

**Writing – original draft:** Sara Alida Volkmer.

**Writing – review & editing:** Susanne Gaube, Martina Raue, Eva Lermer.

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
