## [Decision Letter · Decision Letter 0]

4 Aug 2022

PONE-D-22-17599Troll Story: The Dark Tetrad and Online Trolling Revisited with a Glance at HumorPLOS ONE

Dear Dr. Volkmer,

Thank you for submitting your manuscript to PLOS ONE. After careful consideration, I feel that it has merit but as an initial decision, the reviewers recommend reconsideration of your manuscript following required revisions. Therefore, we invite you to submit a revised version of the manuscript that addresses the points raised during the review process. Please carefully review the attached suggestions of the reviewers.

We look forward to receiving your revised manuscript.

Kind regards,

Yasin Hasan Balcioglu, MD, PhD

Academic Editor

PLOS ONE

Journal Requirements:

Additional Editor Comments:

None

Reviewers' comments:

Reviewer's Responses to Questions

**Comments to the Author**

1. Is the manuscript technically sound, and do the data support the conclusions?

Reviewer #1: Yes

Reviewer #2: Yes

Reviewer #3: Yes

2. Has the statistical analysis been performed appropriately and rigorously? 

Reviewer #1: Yes

Reviewer #2: Yes

Reviewer #3: I Don't Know

3. Have the authors made all data underlying the findings in their manuscript fully available?

Reviewer #1: Yes

Reviewer #2: Yes

Reviewer #3: Yes

4. Is the manuscript presented in an intelligible fashion and written in standard English?

Reviewer #1: Yes

Reviewer #2: Yes

Reviewer #3: Yes

5. Review Comments to the Author

**Reviewer #1:** I appreciate the opportunity to review this pertinent and contemporary article. It is an important topic and aids in filling a most needed void in online challenge literature. The paper's strengths involve using an experimental methodological system that detects the factors underlying online trolling such as personality traits. However, there are some significant and minor issues with the work as presented.

Although the scales' lack of validity and reliability studies stood out as a limitation, Cronbach's alpha values were given. Please indicate that the Humor style questionnaire does/does not have validity and reliability status in German.

Although the discussion of the hypotheses seems sufficient, it has been stated that the correlations between trolling and dark personality traits are contradictory in the literature. Still, this issue has not been discussed. Literature information and comments on this would be good.

**Reviewer #2: **The present article sought to identify the relationship between the Dark Tetrad traits and trolling behavior, considering humor styles. It is possible to see the potential in the article and the author's objective to advance scientifical understanding. However, I have a few concerns regarding the manuscript. I mentioned them by major and minor categories.

Major

- The definition of trolling was not fully explored. In some sections of the manuscript, aggressive behavior (which I understand as one of the factors that make up trolling behavior) is defined separately. Also, the authors use Hardaker's (2010) definition of trolling; while this is not a problem, they suggest in the limitations section other aspects of trolling behavior, which indicates only a superficial definition in the introduction. I suggest reading “The dark side of the Internet – Hannah Barton from An introduction to cyberpsychology (2016)” the authors can find a broader definition of trolling and its subfactors. In all, it is necessary a vertical mapping of the construct.

- The discussion needs improvement. The authors mostly re-state information already presented in the results sections. For example, Machiavellianism and Narcissism were not related to trolling, but why is that? Also, why psychopathy and sadism would be present in trolling behavior. It is necessary to provide a rationale for such results. I usually do not like to suggest my work, but in our article “Considering sadism in the shadow of the Dark Triad traits: A meta-analytic review of the Dark Tetrad – Bonfá-Araujo et al. (2022),” there is a qualitative section where the authors can find more materials regarding the Dark Tetrad traits and online aspects.

- Finally, the authors should provide more information about the adaptation process of the instruments used. This is especially true for the trolling measure that presented low internal consistency. Were different models tested? All measures presented adequate indexes for their German version? These questions need to be explored.

Minor

- Abstracts usually are presented in one single paragraph.

- Keywords could be different from words that already appear on the title to maximize reach once the article is published.

- The article needs grammatical revision. For example, page 16 says Dark Tetra instead of Dark Tetrad. The first paragraph of the discussion also needs revision, the H2 is called fourth, and the H4 is called third in the text.

- The introduction section was chosen to be presented in topics. While this is not a problem, APA suggests that subtopics must have at least two paragraphs. I understood the authors' rationale, but sometimes the separation seems pointless.

**Reviewer #3: **Manuscript: PONE-D-22-17599

Overview: I reviewed the manuscript titled “Troll Story: The Dark Tetrad and Online Trolling Revisited with a Glance at Humor”. Overall, I commend the authors on their well-presented and well-written paper! The study replicates previous findings and extends them by adding new findings about humor styles and trolling behavior. I recommend this paper for publishing but with minor revisions, detailed below:

Introduction:

1. Generally, the introduction is well-resourced and organized clearly.

2. Section 1.2 – While adding a table to show the different correlations between Dark Tetrad traits and trolling behavior is a great visual, it would also benefit the reader to understand why some of these Dark traits may be linked to trolling behaviors. E.g., people with high scores on psychopathy tend to act impulsively and may react to being triggered online…Machiavellianism is associated with hostility and behaviors to regain/maintain control of a situtation..etc). I would suggest adding a quick paragraph to this end.

3. Section 1.3 – I would recommend reorganizing the structure of this section. A suggestion would be to start by defining the different humor styles, and then connecting humor style and trolling behaviors. It would flow better if the link between humor styles and trolling behavior was established before connecting these with Dark personality traits.

4. Citations required for the section below:

Hence, it might also be possible that feeling excluded motivates people to troll other

internet users to avenge themselves. With quick access to social media, trolling posts/comments might be an easy way for people who just experienced social exclusion to regain their perceived control.

Methods:

1. Materials – More a question of curiosity – the Short Dark Triad scale was employed in addition to the Comprehensive Assessment of Sadistic Tendencies. Why was the Short Dark Tetrad scale (Paulhus, Buckels, Trapnell, & Jones, 2020) not used? Additionally, CAST has 18 items dedicated to sadism while the SD3 is a short measure with roughly 9 items measuring psychopathy, Machiavellianism, and narcissism.

As there has been some contention about this in the literature (e.g. Glenn, A. L., & Sellbom, M. (2015). Theoretical and empirical concerns regarding the Dark Triad as a construct. Journal of Personality Disorders, 29, 36–377. https://doi.org/10.1521/pedi_2014_28_162), I would be curious to know if there would have been different nuances had the independent measures of the Dark Triad been used. I would recommend noting this in the discussion.

2. Please proof-read the manuscript as there a few errors (e.g., under 2.3, there is a

bracket after HSQ, and Tetrad is missing a letter in the heading 3.2.1).

3. It may be a good idea to explain what the benefits of a quantile regression are, and what the different quartiles mean.

Results:

1. Remain consistent with hypothesis numbering. E.g., under section 3.2.2, “To test our fourth hypothesis (H2: …)”, would H2 not be your second hypothesis? Similar errors are present in the remainder of the results and discussion section.

Discussion:

1. Great job summing up your findings! The mismatch between hypotheses numbers needs to be corrected in this section too.

2. Good job justifying non-significant results about social exclusion.

3. In the last section of the paper, the link between trolling behavior, humor styles, and the Dark Tetrad traits is mentioned. It would be beneficial to elucidate this link, perhaps in the introduction too, so that the manuscript ties together better. Currently, to me, the research on trolling behavior + humor styles, and trolling behavior + Dark Tetrad looks pretty separate, and it’s unclear why they’re both being studied together.

- To this end, it may be useful to think of the link between DT traits and verbal aggression/hostility.

References:

1. Overall, it looks like a few of the references are quite old. Much like this study, there’s new papers published and if/where possible, it would be great to see slightly more up-to-date citations used.

6. PLOS authors have the option to publish the peer review history of their article (what does this mean?). If published, this will include your full peer review and any attached files.

Reviewer #1: **Yes: **Bahadir Turan

Reviewer #2: **Yes: **Bruno Bonfá-Araujo

Reviewer #3: No

---

## [Author Response · Author response to Decision Letter 0]

17 Sep 2022

We would like to thank the reviewers for their nuanced feedback and have integrated it into our updated manuscript. We believe the article has improved substantially thanks to the reviewers’ comments and hope the manuscript is now publishable. Below, we outline all changes made to the manuscript based on each reviewer’s comments. 

Reviewer #1

Reviewer #1: I appreciate the opportunity to review this pertinent and contemporary article. It is an important topic and aids in filling a most needed void in online challenge literature. The paper's strengths involve using an experimental methodological system that detects the factors underlying online trolling such as personality traits. However, there are some significant and minor issues with the work as presented. 

Response by Authors: We thank Reviewer 1 for seeing merit in the manuscript and have taken the suggested changes into full consideration. 

Reviewer #1: Although the scales' lack of validity and reliability studies stood out as a limitation, Cronbach's alpha values were given. Please indicate that the Humor style questionnaire does/does not have validity and reliability status in German.

Response by Authors: We have added a statement about the lack of an official validation study and added a reference to research with over 1,000 participants that gives some first indication of factorial validity by Ruch and Heintz (2016): 

While no official validation study for the German HSQ exists, there is support for the factorial validity of the German translation we used (Ruch & Heintz, 2016). 

Reviewer #1: Although the discussion of the hypotheses seems sufficient, it has been stated that the correlations between trolling and dark personality traits are contradictory in the literature. Still, this issue has not been discussed. Literature information and comments on this would be good.

Response by Authors: We would like to thank Reviewer #1 for this point. We have consequently expanded our discussion on the topic to explain differences in effects better.

Added discussion points on the Dark Tetrad: 

A recent review indicates (a) that sadism generally motivates trolling more than the remaining Dark Tetrad facets and (b) that the association of sadism with psychopathy is stronger compared to the relationships with narcissism or Machiavellianism (Bonfá-Araujo et al., 2022). The same review also suggests that different aggressive behaviors show stronger associations with sadism and psychopathy but not necessarily with narcissism (Bonfá-Araujo et al., 2022). The finding that sadism is a better predictor of trolling than the Dark Triad facets (Bonfá-Araujo et al., 2022) mirrors our robustness quantile regression where, after low loading items were excluded, only sadism predicted trolling motivation. 

These patterns may be due to the stronger association between sadism and psychopathy with aggressive behaviors, but it may also be due to the often one-dimensional assessment of the Dark Triad facets (Miller et al., 2019). The Dark Triad research has been criticized because measures often neglect the multidimensionality of the Dark Triad facets (Miller et al., 2019). Investigating subcomponents of the Dark Tetrad facets might have provided more nuanced insights into the relationships between personality and trolling behavior. For example, Vize and colleagues (Vize et al., 2019) investigated different facets of narcissism and found that grandiose narcissism was more important in explaining proactive aggression whereas vulnerable narcissism was more important for reactive aggression. In the context of our experiment which manipulated social exclusion, a differentiation between grandiose and vulnerable narcissism might have aided our understanding. 

The concept of the Dark Tetrad has received further criticism: Some researchers suggest that narcissism and Machiavellianism are features of psychopathy and that, thus, the Dark Triad does not explain variance beyond psychopathy (Glenn & Sellbom, 2015). Finally, humor research has indicated that psychopathy outperforms the other facets of the Dark Tetrad in explaining aggressive humor and katagelasticism (Torres-Marín et al., 2022) which may instigate trolling behavior (Brauer et al., 2022). Sadism also uniquely predicts katagelasticism although to a lesser degree than psychopathy (Torres-Marín et al., 2022). In contrast, narcissism is associated with lighter forms of humor that enable relationship-building while Machiavellianism is strongly associated with the use of irony and the fear of being laughed at (Torres-Marín et al., 2022). 

As such, several potential reasons for our findings arise: (a) Psychopathy and sadism are stronger predictors of aggressive behavior, (b) we did not assess the multidimensionality of narcissism, Machiavellianism, and psychopathy, (c) effects of narcissism and Machiavellianism may already be explained by including psychopathy due to an overlap in definitions, and (d) psychopathy and sadism appear to have stronger associations with katagelasticism than narcissism and Machiavellianism. Finally, sadism may outperform psychopathy in our robustness analysis, since people with high sadistic tendencies feel greater aggressive pleasure which may motivate people to behave aggressively (Chester et al., 2019), here: motivation to troll others. This intrinsic enjoyment of inflicting pain is not a component of psychopathy (Hare, 1985).

 

Reviewer #2

Reviewer #2: The present article sought to identify the relationship between the Dark Tetrad traits and trolling behavior, considering humor styles. It is possible to see the potential in the article and the author's objective to advance scientifical understanding. However, I have a few concerns regarding the manuscript. I mentioned them by major and minor categories.

Response by Authors: We would like to thank Reviewer 2 for his constructive feedback and have aimed to include all criticisms into the revised manuscript. 

Major

Reviewer #2: - The definition of trolling was not fully explored. In some sections of the manuscript, aggressive behavior (which I understand as one of the factors that make up trolling behavior) is defined separately. Also, the authors use Hardaker's (2010) definition of trolling; while this is not a problem, they suggest in the limitations section other aspects of trolling behavior, which indicates only a superficial definition in the introduction. I suggest reading “The dark side of the Internet – Hannah Barton from An introduction to cyberpsychology (2016)” the authors can find a broader definition of trolling and its subfactors. In all, it is necessary a vertical mapping of the construct.

Response by Authors: We thank Reviewer #2 for this reference and pointing out this limitation in our trolling definition. We have integrated insights from the book chapter into our definition and relevance section. We have also added information about non-maleficent trolling to ensure a broader conceptualization of trolling. 

What we added to the definition and explanation of trolls: 

Approaches to trolling other internet users with malicious intent can include veering a conversation off-topic as well as being deliberately controversial, offensive or inflammatory (Kunst, 2019). Why are people becoming trolls? One explanation is that the internet can facilitate disinhibition (Cheung et al., 2021; Suler, 2004) which positively predicts cyberaggression (Kurek et al., 2019). According to an early study on trolls (Shachaf & Hara, 2010), users engage with trolling because they are bored, seek attention or revenge, and find it funny to create trouble for platforms and other users. To create the desired disruption, trolls may write messages that are (a) outwardly sincere, (b) deliberately designed to provoke, (c) waste time through fruitless arguments (Herring et al., 2002). At times, the media and scholars conflate trolling with any negative behavior that occurs online, e.g., cyberbullying, parody, or flaming, when the definition of trolling should be limited to social phenomena “performed individually or collectively in varying online contexts, which involves the use of antagonism, deception and vigilantism […] to provoke reactions from people or institutions” (p. 1,078) (Demsar et al., 2021). 

Notably, the above describes a kind of trolling behavior that aims to negatively affect other users and online discussions. However, trolling can occur in a more light-hearted or even amicable way, e.g., Sanfilippo and colleagues (Sanfilippo et al., 2018) differentiate between serious and humorous trolling. This differentiation is also highlighted by the distinction participants drew between circumstantial trolling and trolls who are committed to irritate and iterate their actions (Sanfilippo et al., 2018). Here, we are interested in the malevolent troll as defined by Hardaker (Hardaker, 2010) because of their high relevance to society due to their anti-social behaviors (Sanfilippo et al., 2018)

Reviewer #2: - The discussion needs improvement. The authors mostly re-state information already presented in the results sections. For example, Machiavellianism and Narcissism were not related to trolling, but why is that? Also, why psychopathy and sadism would be present in trolling behavior. It is necessary to provide a rationale for such results. I usually do not like to suggest my work, but in our article “Considering sadism in the shadow of the Dark Triad traits: A meta-analytic review of the Dark Tetrad – Bonfá-Araujo et al. (2022),” there is a qualitative section where the authors can find more materials regarding the Dark Tetrad traits and online aspects.

Response by Authors: Again, we thank Reviewer #2 for this remark and the valuable reference which has been integrated into the paper. We have outlined more research on the Dark Tetrad, including criticisms of it as suggested by Reviewer #3, which can explain the findings. 

Added discussion points on the Dark Tetrad: 

A recent review indicates (a) that sadism generally motivates trolling more than the remaining Dark Tetrad facets and (b) that the association of sadism with psychopathy is stronger compared to the relationships with narcissism or Machiavellianism (Bonfá-Araujo et al., 2022). The same review also suggests that different aggressive behaviors show stronger associations with sadism and psychopathy but not necessarily with narcissism (Bonfá-Araujo et al., 2022). The finding that sadism is a better predictor of trolling than the Dark Triad facets (Bonfá-Araujo et al., 2022) mirrors our robustness quantile regression where, after low loading items were excluded, only sadism predicted trolling motivation. 

These patterns may be due to the stronger association between sadism and psychopathy with aggressive behaviors, but it may also be due to the often one-dimensional assessment of the Dark Triad facets (Miller et al., 2019). The Dark Triad research has been criticized because measures often neglect the multidimensionality of the Dark Triad facets (Miller et al., 2019). Investigating subcomponents of the Dark Tetrad facets might have provided more nuanced insights into the relationships between personality and trolling behavior. For example, Vize and colleagues (Vize et al., 2019) investigated different facets of narcissism and found that grandiose narcissism was more important in explaining proactive aggression whereas vulnerable narcissism was more important for reactive aggression. In the context of our experiment which manipulated social exclusion, a differentiation between grandiose and vulnerable narcissism might have aided our understanding. 

The concept of the Dark Tetrad has received further criticism: Some researchers suggest that narcissism and Machiavellianism are features of psychopathy and that, thus, the Dark Triad does not explain variance beyond psychopathy (Glenn & Sellbom, 2015). Finally, humor research has indicated that psychopathy outperforms the other facets of the Dark Tetrad in explaining aggressive humor and katagelasticism (Torres-Marín et al., 2022) which may instigate trolling behavior (Brauer et al., 2022). Sadism also uniquely predicts katagelasticism although to a lesser degree than psychopathy (Torres-Marín et al., 2022). In contrast, narcissism is associated with lighter forms of humor that enable relationship-building while Machiavellianism is strongly associated with the use of irony and the fear of being laughed at (Torres-Marín et al., 2022). 

As such, several potential reasons for our findings arise: (a) Psychopathy and sadism are stronger predictors of aggressive behavior, (b) we did not assess the multidimensionality of narcissism, Machiavellianism, and psychopathy, (c) effects of narcissism and Machiavellianism may already be explained by including psychopathy due to an overlap in definitions, and (d) psychopathy and sadism appear to have stronger associations with katagelasticism than narcissism and Machiavellianism. Finally, sadism may outperform psychopathy in our robustness analysis, since people with high sadistic tendencies feel greater aggressive pleasure which may motivate people to behave aggressively (Chester et al., 2019), here: motivation to troll others. This intrinsic enjoyment of inflicting pain is not a component of psychopathy (Hare, 1985).

Reviewer #2: - Finally, the authors should provide more information about the adaptation process of the instruments used. This is especially true for the trolling measure that presented low internal consistency. Were different models tested? All measures presented adequate indexes for their German version? These questions need to be explored.

Response by Authors: We have provided more information on the adaptation of the German translation of the GAIT and the development of the IAIT. 

Moreover, we also brought up the point of our translation and use of non-validated measures in the limitations section. 

Prompted by your comment, we conducted confirmatory factor analyses for all measures. Although the overall fits were acceptable, we noticed issues with some low-loading items. Hence, we conducted an additional robustness analysis with means that did not include the low loading items and report it in an Appendix. Overall, the results do not change for correlational and t-test analyses. However, the robustness analysis suggests that sadism is a more important predictor for immediate trolling motivation than psychopathy. We have added a short section about the robustness of our findings at the end of your results section. 

Statement added in Methods: 

The translation process involved several revisions whereupon each revision aimed to maximize the accuracy of the translation while simultaneously maximizing the naturalness of the German formulation. All people involved in this process were fluent in English and German and familiar with the concept of internet trolling. […] The authors created the IAIT measure by reformulating the items to have them address the present moment (e.g., “Just now, I want to share posts/comments that I think are funny, although some people might think they are offensive.”). This process also involved several steps during which the item formulations were clarified and improved. For this adaptation, as with the GAIT, the authors critically investigated the item formulations for understandability and face validity.

Statement added to Limitations: 

Since no official and validated German scales of the GAIT and CAST existed, we used our translation of the scales. While we aimed to capture the original meaning of the items and the concepts to maximize the naturalness of the German formulations, this remains a limitation of the present study.

Minor

Reviewer #2: - Abstracts usually are presented in one single paragraph.

Response by Authors: This has been adjusted, we now present the abstract in a single paragraph.

Reviewer #2: - Keywords could be different from words that already appear on the title to maximize reach once the article is published.

Response by Authors: We thank Reviewer 2 for this kind remark and have adapted the keywords.

Reviewer #2: - The article needs grammatical revision. For example, page 16 says Dark Tetra instead of Dark Tetrad. The first paragraph of the discussion also needs revision, the H2 is called fourth, and the H4 is called third in the text.

Response by Authors: We apologize for these errors and have revised the manuscript. 

Reviewer #2: - The introduction section was chosen to be presented in topics. While this is not a problem, APA suggests that subtopics must have at least two paragraphs. I understood the authors' rationale, but sometimes the separation seems pointless.

Response by Authors: We understand this rational and have adjusted our introduction structure accordingly. 

Our new structure: 

1.1. Online Trolling: Definition and Relevance 

1.2. The Dark Tetrad and Online Trolling 

1.3. Humor Styles and Online Trolling

1.4. Social Exclusion and Online Trolling

Reviewer #3 

Reviewer #3: Manuscript: PONE-D-22-17599

Reviewer #3: Overview: I reviewed the manuscript titled “Troll Story: The Dark Tetrad and Online Trolling Revisited with a Glance at Humor”. Overall, I commend the authors on their well-presented and well-written paper! The study replicates previous findings and extends them by adding new findings about humor styles and trolling behavior. I recommend this paper for publishing but with minor revisions, detailed below:

Response by Authors: We thank Reviewer #3 for their constructive criticism which we have integrated into our updated manuscript. 

Introduction:

1. Generally, the introduction is well-resourced and organized clearly.

Reviewer #3: 2. Section 1.2 – While adding a table to show the different correlations between Dark Tetrad traits and trolling behavior is a great visual, it would also benefit the reader to understand why some of these Dark traits may be linked to trolling behaviors. E.g., people with high scores on psychopathy tend to act impulsively and may react to being triggered online…Machiavellianism is associated with hostility and behaviors to regain/maintain control of a situtation..etc). I would suggest adding a quick paragraph to this end.

Response by Authors: This is a good point which we neglected in the previous manuscript. We have adjusted the section on the Dark Tetrad and trolling to provide more insights about why the constructs may be linked. 

New paragraph on reasons why trolling may be linked to the Dark Tetrad: 

[…] research has confirmed the link between people scoring high on the Dark Tetrad traits and trolling behavior (Buckels et al., 2014, 2019; Lopes & Yu, 2017; March, 2019; March et al., 2017; Paananen & Reichl, 2019; Sest & March, 2017) which may be due to lower affective empathy in these individuals (Blötner et al., 2021; Bonfá-Araujo et al., 2022; Pajevic et al., 2018), a tendency for moral disengagement (Erzi, 2020), and reduced behavioral inhibition anxiety (Thomas & Egan, 2022). Moreover, all four facets are positively associated with dominance (Blötner et al., 2022) and social dominance orientation is also associated with past trolling and acceptance of trolling (Bentley & Cowan, 2021). Another reason for the association between the Dark Tetrad and trolling behavior could be intrinsic enjoyment: Research indicates that sadism, psychopathy, Machiavellianism, and narcissism correlate positively with one’s enjoyment of viewing violent stimuli (Thomas & Egan, 2022). Sadism is also related to experiencing greater pleasure during the aggression (Chester et al., 2019). Psychopathy specifically may also be related to trolling behavior because of its association with impulsivity (Ben-Yaacov & Glicksohn, 2020; Skeem et al., 2011). In the case of narcissism specifically, Vize and colleagues (Vize et al., 2019) showed that antagonism primarily drives aggressive behaviors.

Reviewer #3: 3. Section 1.3 – I would recommend reorganizing the structure of this section. A suggestion would be to start by defining the different humor styles, and then connecting humor style and trolling behaviors. It would flow better if the link between humor styles and trolling behavior was established before connecting these with Dark personality traits.

Response by Authors: We thank Reviewer #3 for this suggestion and have restructured the humor section accordingly to create a better reading flow. 

New structure of this section: 

- Definition of humor styles 

- Trolling and aggressive humor 

- Dark Tetrad and aggressive humor 

Reviewer #3: 4. Citations required for the section below:

Hence, it might also be possible that feeling excluded motivates people to troll other

internet users to avenge themselves. With quick access to social media, trolling posts/comments might be an easy way for people who just experienced social exclusion to regain their perceived control.

Response by Authors: We apologize for the lack of citations and have adjusted the sentence. 

Adjusted sentence with added citations: 

Hence, based on the prior research described above (Chen et al., 2022; Cook et al., 2018; Smith & Williams, 2004), we suggest that it might also be possible that feeling excluded motivates people to troll other internet users to avenge themselves. With quick access to social media or social exclusion occurring online, trolling posts/comments might be an easy way for people who just experienced social exclusion to regain their perceived control (Gerber & Wheeler, 2009; Twenge et al., 2001; Warburton et al., 2006).

Methods:

Reviewer #3: 1. Materials – More a question of curiosity – the Short Dark Triad scale was employed in addition to the Comprehensive Assessment of Sadistic Tendencies. Why was the Short Dark Tetrad scale (Paulhus, Buckels, Trapnell, & Jones, 2020) not used? Additionally, CAST has 18 items dedicated to sadism while the SD3 is a short measure with roughly 9 items measuring psychopathy, Machiavellianism, and narcissism.

Response by Authors: This is a good point to bring up. At the time of pre-registration and data collection (data was collected in 2019), the SD4 was not yet published. We decided to use the CAST as it had been used previously in other trolling studies (e.g., Buckels et al., 2014 & 2019) and because an early idea of the project was to investigate the three subdimensions of sadism (verbal, physical, vicarious) as has been done by Paananen and Reichl (2019).

Reviewer #3: As there has been some contention about this in the literature (e.g. Glenn, A. L., & Sellbom, M. (2015). Theoretical and empirical concerns regarding the Dark Triad as a construct. Journal of Personality Disorders, 29, 36–377. https://doi.org/10.1521/pedi_2014_28_162), I would be curious to know if there would have been different nuances had the independent measures of the Dark Triad been used. I would recommend noting this in the discussion.

Response by Authors: We thank Reviewer #3 for this comment and reference. We have included the criticism of the Dark Tetrad into our discussion section.

Added discussion points on the Dark Tetrad: 

These patterns may be due to the stronger association between sadism and psychopathy with aggressive behaviors, but it may also be due to the often one-dimensional assessment of the Dark Triad facets (Miller et al., 2019). The Dark Triad research has been criticized because measures often neglect the multidimensionality of the Dark Triad facets (Miller et al., 2019). Investigating subcomponents of the Dark Tetrad facets might have provided more nuanced insights into the relationships between personality and trolling behavior. For example, Vize and colleagues (Vize et al., 2019) investigated different facets of narcissism and found that grandiose narcissism was more important in explaining proactive aggression whereas vulnerable narcissism was more important for reactive aggression. In the context of our experiment which manipulated social exclusion, a differentiation between grandiose and vulnerable narcissism might have aided our understanding. 

The concept of the Dark Tetrad has received further criticism: Some researchers suggest that narcissism and Machiavellianism are features of psychopathy and that, thus, the Dark Triad does not explain variance beyond psychopathy (Glenn & Sellbom, 2015). Finally, humor research has indicated that psychopathy outperforms the other facets of the Dark Tetrad in explaining aggressive humor and katagelasticism (Torres-Marín et al., 2022) which may instigate trolling behavior (Brauer et al., 2022). Sadism also uniquely predicts katagelasticism although to a lesser degree than psychopathy (Torres-Marín et al., 2022). In contrast, narcissism is associated with lighter forms of humor that enable relationship-building while Machiavellianism is strongly associated with the use of irony and the fear of being laughed at (Torres-Marín et al., 2022). 

Reviewer #3: 2. Please proof-read the manuscript as there a few errors (e.g., under 2.3, there is a

bracket after HSQ, and Tetrad is missing a letter in the heading 3.2.1).

Response by Authors: We thank Reviewer #3 for their sharp eye and have corrected these mistakes. We have also proof-read the manuscript.

Reviewer #3: 3. It may be a good idea to explain what the benefits of a quantile regression are, and what the different quartiles mean.

Response by Authors: We thank Reviewer #3 for pointing out this lack of clarity and have added an ‘Analysis’ subsection to the Methods section where we provide more details about quantile regression, its benefits and its interpretation. 

Added information: 

Our regression assumption checks showed some violations of the homoscedasticity and normal distribution of errors assumption. Hence, we decided to deviate from our pre-registration and conducted a quantile regression analysis, which allows for residuals to have different variances (Konstantopoulos et al., 2019) and does not assume parametric distributional form (here normal) of the errors (Cade & Noon, 2003). Consequently, a quantile regression analysis should be better suited for our data. Moreover, prior research has compared simple linear regression with quantile regression for personality trait data and concluded that quantile regression can showcase more nuanced and heterogeneous effects (Koenker, 2017; van Zyl & de Bruin, 2018). 

Unlike OLS regression, quantile regression relies on quantiles of the outcome variable which results in several coefficients for a single covariate (Koenker, 2005). These coefficients are interpreted based on their respective quantile of the outcome variable (Koenker, 2005). For example, while OLS regression may indicate an average effect of a woman’s partner’s meanness on relationship satisfaction, quantile regression can show that for the 15th quantile of relationship satisfaction (i.e., the least satisfied women), the partner’s meanness can reduce relationship satisfaction by 0.61 points, while partner’s meanness only reduces relationship satisfaction by 0.14 points for the 85th quantile (i.e., the most satisfied women) (Pilch et al., 2022). 

Results:

Reviewer #3: 1. Remain consistent with hypothesis numbering. E.g., under section 3.2.2, “To test our fourth hypothesis (H2: …)”, would H2 not be your second hypothesis? Similar errors are present in the remainder of the results and discussion section.

Response by Authors: We would like to apologize for these errors and have checked our results and discussion for continuity errors. 

Discussion:

Reviewer #3: 1. Great job summing up your findings! The mismatch between hypotheses numbers needs to be corrected in this section too.

Response by Authors: We have adjusted these errors as well. 

Reviewer #3: 2. Good job justifying non-significant results about social exclusion. 

3. In the last section of the paper, the link between trolling behavior, humor styles, and the Dark Tetrad traits is mentioned. It would be beneficial to elucidate this link, perhaps in the introduction too, so that the manuscript ties together better. Currently, to me, the research on trolling behavior + humor styles, and trolling behavior + Dark Tetrad looks pretty separate, and it’s unclear why they’re both being studied together.

- To this end, it may be useful to think of the link between DT traits and verbal aggression/hostility.

Response by Authors: We go into more depth in the introduction subsection on humor and describe links between the Dark Tetrad and humor. Specifically, we also introduce the concept of katagelasticism (i.e., the joy of laughing at others) and how it is linked to both trolling and the Dark Tetrad.

Added and reformulated information to elucidate the link between the Dark Tetrad, trolling, and humor: 

Indeed, one study (Navarro-Carrillo et al., 2021) recently showed that trolling behavior is associated with aggressive humor as well as katagelasticism (i.e., the joy of laughing at others). Moreover, self-enhancing and self-defeating humor were both related to trolling behavior in that study. Some authors suggest that katagelasticism may be a cause of trolling behavior (Brauer et al., 2022). Additionally, aggressive humor is positively associated with the readiness to be verbally aggressive (Čekrlija et al., 2022) which, we expect, may be expressed in trolling. In fact, sarcasm and mockery can also be one of the tactics trolls use to disrupt discussions (Barton, 2016; Sanfilippo et al., 2018). 

Previous research has also shown that humor styles are associated with the Dark Tetrad. The four traits are linked to inadequate humor (Bonfá-Araujo et al., 2022), e.g., schadenfreude in social, academic, and mourning contexts (Erzi, 2020; Lee, 2019). Humor research indicates that the Dark Tetrad facets are linked to how people utilize and enjoy humor (Torres-Marín et al., 2022), e.g., aggressive humor largely overlaps with the comic style sarcasm (Heintz & Ruch, 2019) and higher Machiavellianism as well as subclinical psychopathy scores are associated with aggressive humor (Martin et al., 2012; Veselka et al., 2010). More specifically, psychopathy appears to have a stronger connection with aggressive humor and katagelasticism than Machiavellianism and narcissism (Torres-Marín et al., 2022). Sadism also uniquely explains variance in katagelasticism beyond the Dark Triad (Torres-Marín et al., 2022). Importantly, katagelasticism not only involves enjoying laughing at others but also actively seeking out situations where one can ridicule others (Torres-Marín et al., 2022). We believe that internet trolling may be an expression of katagelasticism in people who score highly on the Dark Tetrad facets. In sum, it is likely that trolling behavior is also associated with more aggressive humor and this link may exist because people who score highly on the Dark Tetrad use humor differently than people who score lower on the Dark Tetrad.

References:

Reviewer #3: 1. Overall, it looks like a few of the references are quite old. Much like this study, there’s new papers published and if/where possible, it would be great to see slightly more up-to-date citations used.

Response by Authors: We have updated the references and the reference section by screening for newer publications on the topics and constructs. In doing so, we have added 40 references. We have integrated these into the manuscript while keeping some of the older, formative papers of trolling research (e.g., Buckels et al., 2014).

---

## [Decision Letter · Decision Letter 1]

29 Nov 2022

PONE-D-22-17599R1Troll Story: The Dark Tetrad and Online Trolling Revisited with a Glance at HumorPLOS ONE

Dear Dr. Volkmer,

Thank you for submitting your manuscript to PLOS ONE. After careful consideration, we feel that it has merit but does not fully meet PLOS ONE’s publication criteria as it currently stands. Therefore, we invite you to submit a revised version of the manuscript that addresses the points raised during the review process.

We look forward to receiving your revised manuscript.

Kind regards,

Yasin Hasan Balcioglu, MD, PhD

Academic Editor

PLOS ONE

Journal Requirements:

Reviewers' comments:

Reviewer's Responses to Questions

**Comments to the Author**

1. If the authors have adequately addressed your comments raised in a previous round of review and you feel that this manuscript is now acceptable for publication, you may indicate that here to bypass the “Comments to the Author” section, enter your conflict of interest statement in the “Confidential to Editor” section, and submit your "Accept" recommendation.

Reviewer #1: All comments have been addressed

Reviewer #2: All comments have been addressed

Reviewer #3: All comments have been addressed

2. Is the manuscript technically sound, and do the data support the conclusions?

Reviewer #1: Yes

Reviewer #2: Yes

Reviewer #3: Yes

3. Has the statistical analysis been performed appropriately and rigorously? 

Reviewer #1: Yes

Reviewer #2: Yes

Reviewer #3: Yes

4. Have the authors made all data underlying the findings in their manuscript fully available?

Reviewer #1: Yes

Reviewer #2: Yes

Reviewer #3: Yes

5. Is the manuscript presented in an intelligible fashion and written in standard English?

Reviewer #1: Yes

Reviewer #2: Yes

Reviewer #3: Yes

6. Review Comments to the Author

Reviewer #1: Thank you for the opportunity to re-review manuscript PONE-D-22-17599R1, entitled "Troll Story: The Dark Tetrad and Online Trolling Revisited with a Glance at Humor". The paper could be published as it is.

Best,

Reviewer #2: The manuscript vastly improved. The authors were able to fully address all comments made by the reviewers. I do not have any further suggestions.

Reviewer #3: I would like to congratulate the authors on a splendid reworking of their article! All my comments (and more!) have been satisfactorily addressed and I am pleased to recommend this paper for publishing.

The section on katagelasticism was particularly enjoyable. The intention behind studying the Dark Tetrad traits, trolling, and humour styles together is a lot clearer. Thank you for addressing concerns about the measures that were used.

The only minor error I picked up on was under Section 3.4 “the first graph in Figure 1 shows how Machiavellianism…”. It looks like it would be the either the first row or third graph depicting Machiavellianism. The same section (3.4) has its title in bold, which should be reformatted to normal.

Once again, thank you for taking the time to comb through the manuscript, making all the necessary changes, and adding in more recent citations.

7. PLOS authors have the option to publish the peer review history of their article (what does this mean?). If published, this will include your full peer review and any attached files.

Reviewer #1: **Yes: **Bahadir Turan

Reviewer #2: **Yes: **Bruno Bonfá-Araujo

Reviewer #3: No

---

## [Author Response · Author response to Decision Letter 1]

30 Nov 2022

Reviewer #3: The only minor error I picked up on was under Section 3.4 “the first graph in Figure 1 shows how Machiavellianism…”. It looks like it would be the either the first row or third graph depicting Machiavellianism. The same section (3.4) has its title in bold, which should be reformatted to normal. 

Response by authors: We would like to thank Reviewer #3 for their kind feedback and keen eye. The section heading of 3.4. has been reformatted to normal. We have also fixed the error of referring to the Machiavellianism graph: “the third graph in the first row of Figure 1 shows how Machiavellianism …”

---

## [Decision Letter · Decision Letter 2]

26 Dec 2022

Troll Story: The Dark Tetrad and Online Trolling Revisited with a Glance at Humor

PONE-D-22-17599R2

Dear Dr. Volkmer,

We’re pleased to inform you that your manuscript has been judged scientifically suitable for publication and will be formally accepted for publication once it meets all outstanding technical requirements.

Kind regards,

Yasin Hasan Balcioglu, MD, PhD

Academic Editor

PLOS ONE
---

## [Editor Report · Acceptance letter]

9 Jan 2023

PONE-D-22-17599R2 

Troll Story: The Dark Tetrad and Online Trolling Revisited with a Glance at Humor 

Dear Dr. Volkmer:

I'm pleased to inform you that your manuscript has been deemed suitable for publication in PLOS ONE. Congratulations! Your manuscript is now with our production department. 

Kind regards, 

on behalf of

Dr. Yasin Hasan Balcioglu 

Academic Editor

PLOS ONE